

# An extraction method for nitrogen isotope measurement of ammonium in low concentrated environment

Alexis Lamothe[1], Joel Savarino[1], Patrick Ginot[1], Lison Soussaintjean[2], Elsa Gautier[1], Pete D. Akers[3], Nicolas Caillon[1], Joseph Erbland[1]

[1]Univ. Grenoble Alpes, CNRS, IRD, INRAE, Grenoble INP*, IGE, 38000 Grenoble, France
*Institute of Engineering and Management Univ. Grenoble Alpes
[2]Climate and Environmental Physics, Physics Institute & Oeschger Centre for Climate Change Research, University of Bern, 3012 Bern, Switzerland
[3]Discipline of Geography, School of Natural Sciences, Trinity College Dublin, Dublin, Ireland

10    *Correspondence to*: Alexis Lamothe (alexis.lamothe@univ-grenoble-alpes.fr) or Joel Savarino (joel.savarino@cnrs.fr)

**Abstract.** Ammonia ($NH_3$) participates in nucleation and growth of aerosols and thus plays a major role in atmospheric transparency, pollution, health, and climate related issues. Understanding its emission sources through nitrogen stable isotopes is therefore a major focus of current work to mitigate the adverse effects of aerosol formation. Since ice cores can preserve the past chemical composition of the atmosphere for centuries, they are a top tool of choice for understanding past $NH_3$ emissions through ammonium ($NH_4^+$), the form of $NH_3$ archived in ice. However, the remote or high-altitude sites where glaciers and ice sheets are typically localized have relatively low fluxes of atmospheric $NH_4^+$ deposition which makes ice core samples very sensitive to laboratory $NH_3$ contamination. As a result, accurate techniques for identifying and tracking $NH_3$ emissions through concentration and isotopic measurements are highly sought to constrain uncertainties in $NH_3$ emission inventories and atmospheric reactivity unknowns. Here, we describe a solid phase extraction method for $NH_4^+$ samples of low concentration that limits external contamination and produces precise isotopic results. By limiting $NH_{3atm}$ exposure with a scavenging fume hood and concentrating the targeted $NH_4^+$ through ion exchange resin, we successfully achieve isotopic analysis of 50 nmol $NH_4^+$ samples with a 0.6 ‰ standard deviation. This extraction method is applied to an alpine glacier ice core from Col Du Dôme, Mont-Blanc, where we successfully demonstrate the analytical approach through the analysis of two replicate 8 m water equivalent ice cores representing 4 years of accumulation with a reproducibility of ± 2.1 ‰. Applying this methodology to other ice cores in alpine and polar environments will open new opportunities for understanding past changes in $NH_3$ emissions and atmospheric chemistry.

## 1 Introduction

Ammonia ($NH_3$) is an important compound of the nitrogen cycle that has many direct and indirect effects on public health and the environment. Not only does $NH_3$ react with sulphate and nitrate to form secondary aerosols that adversely affect human health through air pollution (Lelieveld and Pöschl, 2017), but it also impacts soil acidification (Velthof et al., 2011) and changes radiative forcing (Szopa et al., 2021). The main sources of $NH_3$ are assumed to be livestock waste and fertilisers, but considerable proportions are emitted by biomass burning, industry (Sutton et al., 2013), and road traffic through 3-catalyst



engines (Perrino et al., 2012; Reche et al., 2015). Despite these major impacts, quantified inventories of $NH_3$ mass and flux in the environment are highly approximated, due to a lack of a $NH_3$ monitoring network (Fortems-Cheiney et al., 2016) and large

uncertainties in emission factors (to ± 300 %, Fortems-Cheiney et al., 2016), and major discrepancies therefore exist between these inventories and satellite observations and their subsequent top-down inventories (e.g., Zhan et al., 2021; Fortems-Cheiney et al., 2022; Van Damme et al., 2018). These factors result in $NH_3$ inventory uncertainties that cannot be well estimated (European Environment Agency, 2021). Tracking past changes in $NH_3$ is even more difficult because $NH_3$ readily reacts in the environment and is generally not preserved in natural archives. At a time when reducing $NH_3$ emissions is an increasingly

important issue both in academia (Gu et al., 2021; Erisman, 2021) and in politics (European Council, 2016), an improved method of accounting for $NH_3$ emission sources as well as tracking their changes in the past is required.

Despite the relative scarcity of $NH_3$ analytical possibilities, ammonium ($NH_4^+$) can serve as a proxy of $NH_3$ as it is a generally more stable compound formed from the protonation of $NH_3$, and it is known to be preserved in archives such as ice cores. $NH_3$ protonation (Reaction 1) is central in aerosol chemistry through its participation in nucleation and particle growth

in the presence of acids (often $H_2SO_4$ or $HNO_3$) (Kirkby et al., 2011; Wang et al., 2022).

$$NH_3 + H^+ \leftrightarrow NH_4^+, \text{pKa} = 9.2 \tag{R. 1}$$

Stable isotope analyses have become increasingly relevant for determining emission sources of $NH_3$ and other environmental compounds. Because one isotope of an element (e.g., $^{15}N$) is often favoured when a compound (e.g., $NH_4^+$) undergoes physical transformations or chemical reactions, the isotopic ratios of a compound can differ based on the specific history of its formation

and transportation. In this manner, nitrogen isotope ratios ($\delta^{15}N$ defined in Equation 1 and reported relative to standard atmospheric $N_2$), are a means of studying the sources and reactive history of $NH_4^+$, and thus, $NH_3$.

$$\delta^{15}N = \frac{\left(^{15}N/_{14}N\right)_{sample}}{\left(^{15}N/_{14}N\right)_{N_2-air}} - 1 \tag{Eq. 1}$$

Glacial ice is an excellent natural archive for $NH_4^+$ as aerosols are continuously deposited and buried with snow on the glacier surface and then preserved within the ice over long periods of time. By drilling and sequentially melting ice cores, one can

reconstruct the history of changes to these aerosols at local and global scales (e.g., Preunkert et al., 2001; Gautier et al., 2019). The nitrogen isotopic ratios of $NH_4^+$ ($\delta^{15}N(NH_4^+)$) preserved in ice cores should provide a valuable insight into past $NH_4^+$ variability in sourcing and formation, but such a record has not yet been described in the scientific literature.

$NH_4^+$ extraction for nitrogen isotopic analysis outside of ice core research has a long history of investigation. Analysis through an Isotopic Ratio Mass Spectrometer (IRMS) requires the conversion of $NH_4^+$ to a gaseous species, and Rittenberg was among

the first to present an extraction method using Rittenberg oxidation to produce $N_2$ (Wilson et al., 1946). The subsequent development of elemental analyser combustion (Burke et al., 1990) provided an alternative method to produce $N_2$, but both these methods require large sample sizes (µmol) unsuitable for the very low concentrations of $NH_4^+$ in ice core samples (µmol $kg^{-1}$ range). Although later developments in distillation and diffusion techniques offered additional options, substantial analytical limitations remained. $NH_3$ distillation (Preston et al., 1996) is time-consuming, technically demanding, and has been



reported as being sensitive to external contamination (Liu et al., 2014) whilethe diffusion method (O'Deen and Porter, 1980) presents the same drawbacks, in addition to being unreliable at low concentrations.

To overcome these analytical obstacles, a methodology based on ion exchange resin offers more promising results by concentrating $NH_4^+$ from large quantities of ice in a manner similar to that successfully used for analysing the nitrogen isotopes of nitrate in ice cores (Silva et al., 2000; Frey et al., 2009; Erbland et al., 2013). The ion exchange resin can be used in two

ways. First, $NH_4^+$ concentrated on the resin can be followed by immediate combustion to produce $N_2$ (Lehmann et al., 2001), but the required sample size for this approach (100 µmol) makes it unsuitable for many ice core studies. A second technique consists of eluting the resin-concentrated $NH_4^+$, and the resulting aqueous $NH_4^+$ sample can be first converted into $N_2O$ and then into $N_2$ for the IRMS measurement (Zhang et al., 2007). This conversion is based on $NH_4^+$ oxidation to $NO_2^-$ by a hypobromite solution, with $NO_2^-$ then reduced to $N_2O$ by a sodium azide/acetic acid solution. This second method has been

applied to aerosol samples (Kawashima et al., 2021) but has not yet been applied to ice core analyses, likely due to challenges related to the smaller $NH_4^+$ concentrations in ice.

This paper presents an analytical method for the collection and analysis of $NH_4^+$ in small sample sizes (<50 nmol) suitable for ice cores which is then applied to ice from the Col Du Dôme glacier (CDD) on the Mont Blanc massif (4250 m asl, 45.8421°N, 6.8474°E). This $NH_4^+$ analysis is of particular interest because Col Du Dôme ice cores have been found previously to accurately

preserve atmospheric composition on European and local scales (Preunkert et al., 2001, 2003; Maupetit et al., 1995). For our proposed method, contamination control and sample collection were meticulously tested to determine best practices to preserve sample purity. Reproducibility and repeatability of this method are presented for 8 meters water equivalent (m w.e.) of a Col Du Dôme ice core which shows the new potential that this method offers for understanding $NH_4^+$ sources, transport, and reactivity.

**2 Overview of the protocol development**

This section gathers the details to perform an analysis presented in the paper for an ice core sample assuming that the concentration profile is already determined. Figure 1 summarises the main steps – i.e., sample preparation, $NH_4^+$ resin concentration and elution, isotopic measurement, and blank correction – of $\delta^{15}N(NH_4^+)$ measurement in a 10 cm diameter ice core with, for each step, the tests that were performed to develop the methodology and the final methodology. All terms are

precisely defined in the following sections.



# PROTOCOL DEVELOPMENT

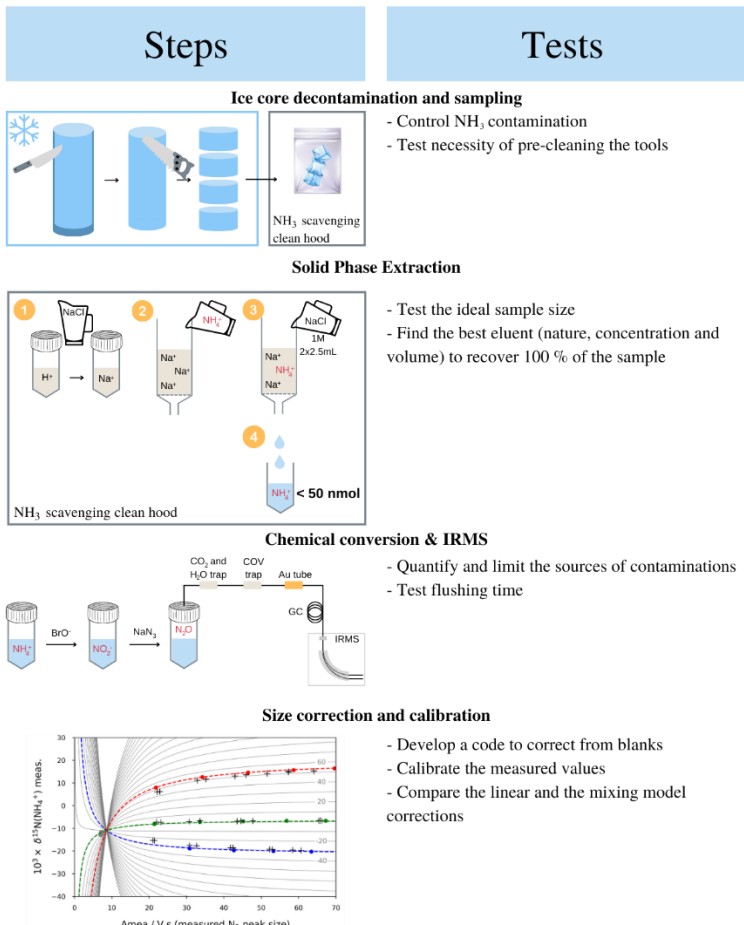

**Figure 1:** Graphical summary of the steps involved in our proposed ice core $NH_4^+$ isotopic analysis, and the tests undertaken at each step to refine the protocol.

Sample preparation:

In a cold room, the first step is to cut the samples. For this, it is necessary to use clean tools that have been decontaminated with MQ ice. The ice core is decontaminated by removing the outermost 2 mm of the external faces. The core is then sampled according to the desired sample size; in our case, with a 50 nmol target, sampling every 25 cm was sufficient. Decontaminated ice samples are sealed in clean plastic bags (Whirl-Pak) and then left to melt within the bags under the $NH_3$ scavenging clean hood.

100        Concentration and elution step:

For each analytical batch, a minimum of 2 blanks, 5 x 3 standards (20, 40, 60, 80, and 100 nmol of the international isotopic standards IAEA-N-1, USGS 25, and USGS 26) and the samples studied must be passed through the cationic resins. The AG 50W that is originally in $H^+$ form must be converted to the $Na^+$ form by preparing a solution of the resin with NaCl 1 M (1:1





m/m). Each extraction column then uses 0.6 mL of this NaCl-resin solution. To begin an $NH_4^+$ extraction, 10 mL of 1 M NaCl
solution is passed through the resin in the column and followed with five successive rinses with 10 mL of MQ water. The resin
is dried by blowing 10 mL of air with a pipette, and the sample (or standard or blank) is loaded by injection at the top of the
extraction column. As the sample passes gravitationally through the resin, the $NH_4^+$ is trapped in the resin and the passed liquid
discarded. Next, the now trapped $NH_4^+$ is eluted into a vial with the injection of 2x2.5 mL of 1 M NaCl at the top of the column
and allowing it to gravitationally pass through the resin. After dripping has completed, the vial containing the $NH_4^+$ in 5 mL 1
M NaCl is sealed. Used resin can be regenerated for reuse by flushing with 10 mL of 1 M NaCl and 5x10 mL of MQ water
and also stored in a freezer for future use.

Chemistry conversion and IRMS analysis:

The measurement of $\delta^{15}N(NH_4^+)$ is performed according to methods described in Zhang et al. (2007) and McIlvin & Altabet
(2005). Hypobromite and the sodium arsenate solutions are prepared under the $NH_3$ scavenging clean hood. The sodium
azide/acetic acid solution is flushed with He at 40 mL min$^{-1}$ for 30 minutes. An autosampler can be used for precise and
repetitive chemistry mixing. We remind that this method uses toxic chemicals that need to be handled respecting strict safety
rules with personal protective equipment wearing. The $NH_4^+$ converted into $N_2O$ then follows the Kaiser et al. (2007) protocol
for the measurements of $\delta^{15}N(NH_4^+)$ with a Finnigan MAT-253 IRMS.

Blank correction:

The calibration and the correction from the blank are performed with the Python code available as a separate link (Lamothe et
al., 2023). This correction determines the best parameters to limit the difference between a theoretical $\delta^{15}N$ based on 15N
mass-balance and the measured $\delta^{15}N$. This allows to calculate calibrated $\delta^{15}N$ values for each standard which can then be used
to correct the sample values.

## 3 Method

### 3.1 Continuous flow analysis $NH_4^+$ concentration measurement

The $NH_4^+$ concentration profile of an ice core is needed to both plan out sampling intervals that contain the desired amount of
$NH_4^+$ and to provide geochemical interpretation. For our analyses, we determined the $NH_4^+$ concentrations with the continuous
flow analysis (CFA) PANDA platform developed at IGE in Grenoble, France (https://panda.osug.fr/). This platform
continuously melts and analyses ice sticks (cross-section: 34x34 mm²) with an online colorimetry module that can measure
$NH_4^+$ concentration as part of its analytical platform. The methodology for measuring $NH_4^+$ concentration through online
colorimetry is described in detail in existing literature (Sigg et al., 1994; Röthlisberger et al., 2000; Kaufmann et al., 2008;
Bigler et al., 2011). In short, soluble $NH_4^+$ from a melted ice core reacts through the Roth ternary reaction with a reducing
phtaldialdehyde (OPA) reagent and sodium sulphite buffered at pH = 9.5. The reaction rate is controlled at 75 °C and the
product creates a highly fluorescent signal at 430 nm (Genfa and Dasgupta, 1989). Three concentration standards at 25, 122,





and 521 ppb were used for daily calibration. The overall system is suitable for ice core measurements with a limit of detection
of $(0.22 \pm 0.03)$ ppb and a linear calibration down to 1 ppm.

## 3.2 Contamination control and ice core cutting

For any sample with low $NH_4^+$ concentrations, atmospheric $NH_3$ present during sample processing can be a substantial source
of contamination. For example, $NH_4^+$ concentrations in alpine ice cores are typically in the $ng.g^{-1}$ (or ppb) to $\mu g.g^{-1}$ (or ppm)
range, and Preunkert et al. reported mean values in the Col Du Dôme ice of $13 \pm 9$ ppb in winter and $180 \pm 58$ ppb in summer
(Preunkert et al., 2000). To determine the potential scale of atmospheric contamination within the laboratory, MQ-water
samples $(18.2\ M\Omega.cm)$ were left open for several hours in a standard extraction fume-hood. Identical experiments were carried
out in a laminar flow purification hood equipped with an activated charcoal filter for $NH_3$ scavenging (ASTM-012 filter on
Iniflow-LAB1050 hood, Initioshop). $NH_4^+$ concentrations for these contamination tests were measured through ion
chromatography (IC).

In addition to contamination of aqueous samples, the physical handling and processing of ice can have several potential
opportunities for $NH_4^+$ contamination. To investigate this route of contamination, we created artificial ice sticks made from
MQ-pure water and replicated our typical ice core cutting and handling process. The MQ ice sticks were decontaminated by
removing the outermost 2 mm of the external faces with a clean ceramic knife. To avoid confusion with other contamination,
we will call this step surface-decontamination. The surface-decontaminated ice sticks were then cut into individual ice samples
using the saws typically used for ice core processing. We also tested the need of pre-cleaning the tools, i.e., the ceramic knife
and the two saws, before processing ice cores. Pre-cleaning the tools consists in scratching all their faces against MQ ice for
about 10 seconds. For this test, one MQ ice stick was surface-decontaminated using tools that had previously been pre-cleaned
with MQ ice, and another MQ ice stick was surface-decontaminated with non-precleaned tools. The resulting ice samples were
sealed in clean plastic bags (Whirl-Pak) and then left to melt within the bags under the $NH_3$ scavenging clean hood. IC
measurements was performed for assessing the $NH_4^+$ contamination levels resulting from the ice core cutting. Contamination
was also a concern during the $NH_4^+$ conversion to $N_2O$ processes, and our efforts to determine potential contamination risk for
these steps will be discussed along with the conversion methodology (section 3.5).

## 3.3 Sample size

Choosing a sample size for $NH_4^+$ isotopic analysis is a compromise between being too large for a practical ice core study and
being so small that the blank fraction's $NH_4^+$ content – i.e., the proportion of the blank size to the sample size – substantially
affects the isotopic and concentration uncertainties. To determine the ideal sample size, we targeted an initial 20–100 nmol
range based on successful past experience with nitrate resin extraction and analysis (Silva et al., 2000; Erbland et al., 2013)
and some preliminary testing. Tests were conducted using the three international $NH_4^+$ isotopic standards IAEA-N-1
$((NH_4)_2SO_4,\ \delta^{15}N = (+0.43 \pm 0.07)$ ‰), USGS 25 $((NH_4)_2SO_4,\ \delta^{15}N = (-30.41 \pm 0.27)$ ‰), and USGS 26 $((NH_4)_2SO_4,\ \delta^{15}N =$
$(+53.75 \pm 0.24)$ ‰), which are referred throughout here as standards J, K, and L, respectively.



Known molar quantities (20, 40, 60, 80, and 100 nmol) of each standard were passed through a cationic exchange resin, and $\delta^{15}N$ values of each standard were determined at each molar quantity. We identified the best suitable sample size as the smallest size that has no substantial isotopic effect from the blank. In practice, this means finding the smallest molar quantity for which the blank fraction does not exceed one third.

**3.4 Elution: define the nature of the eluent, its concentration, and the volume needed**

For $NH_4^+$ capture, we used a solution of cationic resin AG 50W Biorad resin (hydrogen form) in MQ-water (1:1 m/m). To activate and clean the resin, 10 mL of a 1 M eluent are passed through the resin, followed by 5 rinses with 10 mL of MQ-water. The choice of the eluent is discussed below. The resin was dried before and after passing samples through the resin by blowing 10 mL of air with a pipette. This step is not required for Solid Phase Extraction (SPE), but as it did not affect the result, we arbitrarily decided to systematically dry all resins in order to limit all the dead-volumes. The volume of resin used for each $NH_4^+$ concentration must fully capture the entire $NH_4^+$ content of an ice sample to avoid possible isotopic fractionation issues. For this, a sufficient volume of resin must be used that considers the quantity of active sites in the resin in relation to the quantity of ions captured. All concentration and elution steps are fully carried out under the $NH_3$ scavenging purification hood to limit external contamination. Our three isotopic $NH_4^+$ standards were processed at 50 nmol each on three resin volumes of 0.3 mL, 0.6 mL, and 1.2 mL. After elution, $NH_4^+$ was converted into $N_2O$ as described in the next section, and the isotopic and size deviations were determined on the IRMS. Note that it is not possible to directly measure the concentration of $NH_4^+$ in the elution solution due to the high concentration of the ionic eluent, therefore concentration measurements after elution are also done by reading the sample size in IRMS.

We also ran a series of tests to determine which eluent compound and concentration was most effective in eluting the small volumes of ice core-sourced $NH_4^+$ trapped in our resin columns. To elute an ion from an ionic resin, one should take into account the counterion selectivity compared to the studied cation selectivity (here, $NH_4^+$ with a selectivity listed as 1.95). For our BioRad AG50W resin, $H^+$ and $Na^+$ have a relative selectivity of 1.0 and 1.5, respectively. Consequently, we tested the easily sourced NaCl and HCl as possible eluents. KOH was also tested as an alkaline eluent to explore impacts from pH variations on our method. To determine the best eluent for $NH_4^+$ trapped on the cationic resin, 50 nmol of $NH_4^+$ trapped on the resin was each eluted with 2x2.5 ml of 0.1 M NaCl, HCl, or KOH. This eluted $NH_4^+$ was then converted into $N_2O$, and the yield compared to a 50 nmol $NH_4^+$ that was not passed through the cationic resin but instead directly converted to $N_2O$. After setting aside the choice of KOH as the eluent (see details in section 4.3), a second test was performed between NaCl and HCl by increasing their concentration to 1 M.

Elution efficiency was also tested by collecting elution fractions whose sizes were measured by IRMS. Starting with an eluent concentration of 0.1 M, we attempted to extract the isotopic international standards previously mentioned in a 30-100 nmol size range with a volume of 5 mL. Samples and eluent are gravitationally passed through the resin. After elution is completed, the resin is dried again to be consistent with sample loading as mentioned above, $NH_4^+$ sample is sealed with a septum in a



glass vial, and the column can be reprocessed for future use. We found out that one bed of resin can be used at least 25 times

without affecting its effectiveness.

### 3.5 $NH_4^+$ conversion and isotopic measurement

The measurement of $\delta^{15}N(NH_4^+)$ is performed according to the methods described by Zhang et al. (2007) and McIlvin & Altabet (2005). In order to reach the nmol-range size target needed for ice core studies, contamination controls must be performed. For the $NH_4^+$ oxidation, the hypobromite solution preparation can suffer from atmospheric $NH_3$ contamination, but

this contamination is limited by preparation under the previously mentioned $NH_3$ scavenging hood. On the other hand, sodium azide, used as a reagent during the reduction step, is a highly reactive solid which can lead to the formation of nitrite with its oxidation. We first tested the nitrite contamination of the $NaN_3$ reagent by using a Griess colorimetric dosage (e.g., Röthlisberger et al., 2000). It has been reported that flushing the reagent solution with helium for 10 minutes will limit possible nitrite contamination, and so we tested helium flushes of 0 min, 10 min, and 30 min to determine blank sizes of the overall

method.

In the $NH_4^+$ isotopic analysis, samples and blanks undergo the identical analytical process as a suite of standards. Our three international $NH_4^+$ standards J, K, and L were then used to correct for any isotopic effects resulting from processing and analysis. This was used as part of the development of the method for fractionation considerations and also, once the method is validated, for sample calibration. Measurements of $\delta^{15}N(NH_4^+)$ were performed on a Finnigan MAT-253 isotope ratio mass-

spectrometer (IRMS) following the protocol detailed by Kaiser et al. (2007). Briefly, $N_2O$ is decomposed into $N_2$ and $O_2$ that are then separated on a gas column prior to analysis by the IRMS.

### 3.6 Isotopic corrections

Since some degree of background contamination is inevitable, a blank is used to quantify the level of this contamination. As the blank size is not null, the sample $NH_4^+$ mixes with the blank $NH_4^+$ before its conversion into $NO_2^-$. Additionally, only one

atom of N in $N_2O$ comes from $NO_2^-$ while the second one originates from the azide reagent (Zhang et al., 2007).

$$A_{measured} = A_{blank} + A_{azide} + A_{sample} \qquad \text{(Eq. 2)}$$

where $A_{measured}$, $A_{blank}$, $A_{azide}$, and $A_{sample}$ respectively represent the measured amount, the blank amount from contamination, the chemistry reaction blank amount from azide, and the sample (or standard) amount. Equation 2 can immediately be expressed as follow since $A_{azide} = A_{measured}/2$ (Figure S1):

$$A_{sample} = \frac{A_{measured}}{2} - A_{blank} \qquad \text{(Eq. 3)}$$

Thus, we can develop an isotopic mass balance on nitrogen in the final $N_2O$ measured on IRMS (details in Figure S1). We assume each step – i.e., SPE, $NH_4^+$ oxidation by $BrO^-$, and $NO_2^-$ reduction by $NaN_3$ – to be total (i.e., all the $NO_2^-$ is consumed).

$$A_{measured} \times \delta^{15}N_{measured} = A_{blank} \times \delta^{15}N_{blank} + A_{azide} \times \delta^{15}N_{azide} + A_{sample} \times \delta^{15}N_{sample} \qquad \text{(Eq. 4)}$$



where $\delta^{15}N_{measured}$, $\delta^{15}N_{blank}$, $\delta^{15}N_{azide}$, and $\delta^{15}N_{sample}$ respectively represent the $^{15}N$ isotopic composition measured, from the
blank, from the azide reagent, and from the sample. In Equation 4, three unknowns stay: $A_{blank}$, $\delta^{15}N_{blank}$, and $\delta^{15}N_{azide}$. The
goal of the developed Python code is to determine this tuple, assuming it to be constant on all the samples of one batch. This
code accounts for both a 'size correction', i.e., the relative contribution of blank to the sample size, and also for any isotopic
fractionation that occurs during the $N_2O$ measurement through the "gold method" conversion of $N_2O$ into $N_2$ (Kaiser et al.,
2007). To do so, we first vary the tuple of unknowns into a range of possible values, then calculate a theoretical $\delta^{15}N_{measured}$
value, and finally correct the $\delta^{15}N_{measured}$ values from the best tuple determined as the smallest residual difference between
$\delta^{15}N_{scc}$ (size corrected and calibrated) and $\delta^{15}N_{measured}$. This code is performed with 4 main steps that are described here after:

1.    For the international isotopic standards – IAEA-N-1, USGS 25, and USGS 26 respectively noted J, K, and L – ranging
different sizes, the code calculates the $\delta^{15}N_{measured}$ value that should be theoretically obtained and calculated with
Equation 5 based on the J, K, and L isotopic values from literature (i.e., +0.43 ‰, –30.41 ‰, and +53.75 ‰). $A_{blank}$,
$\delta^{15}N_{blank}$, $\delta^{15}N_{azide}$ vary in wide – yet realistic – intervals.

$$\delta^{15}N_{measured} = \delta^{15}N_{sample}\frac{A_{measured}-2\times A_{blank}}{2\times A_{measured}} + \frac{A_{blank}}{A_{measured}}\delta^{15}N_{blank} + \frac{\delta^{15}N_{azide}}{2} \qquad \text{(Eq. 5)}$$

2.    The second step consists in comparing the theoretically obtained $\delta^{15}N_{measured}$ values with the empirically measured
$\delta^{15}N_{measured}$ values from the mass spectrometer. This accounts for the isotopic fractionation occurring during the
$N_2O \rightarrow N_2$ conversion.

3.    The obtained linear relationship between the theoretical $\delta^{15}N_{measured}$ and the observed $\delta^{15}N_{measured}$ is used to calculate
calibrated $\delta^{15}N$ values for each standard.

4.    From these calibrated standard values, Equation 5 can be reversed to obtain $\delta^{15}N_{scc}$ (size corrected and calibrated) as
shown in Equation 6. A residual error is calculated as the standard deviation of the difference between the calculated
$\delta^{15}N_{scc}$ values and the international reference $\delta^{15}N$ values. The best tuple is associated with the smallest residual error.

$$\delta^{15}N_{scc} = \frac{A_{measured}\times\left(\delta^{15}N_{calibrated}-\frac{\delta^{15}N_{azide}}{2}\right)-A_{blank}\times\delta^{15}N_{blank}}{\frac{A_{measured}}{2}-A_{blank}} \qquad \text{(Eq. 6)}$$

By comparing the calculated and measured blank isotopic compositions and sizes, one can check that the isotopic composition
converges towards the blank with smaller sizes. Once the tuple is fixed, the mixing model is again reversed to correct and
calibrate the samples. The Python scripts used for this correction are available as a separate link (Lamothe et al., 2023). This
general correction which considers blank size and calibration will hereafter be referred to as the mixing model correction.

**3.7 Application on a Col Du Dôme ice core and reproducibility**

As part of the Ice Memory programme, a 126 m ice core (CDD16-C01) was drilled in 2016 on the CDD glacier of Mont-Blanc
at 4250 m asl. The CDD glacier is a high-accumulation glacier (from 0.5 to 2.4 m w.e.yr$^{-1}$) on which a fair amount of literature
has been published (e.g., Legrand et al., 2021; Preunkert et al., 2019; Guilhermet et al., 2013; Preunkert et al., 2003, 2001;
Maupetit et al., 1995), largely dedicated to the dynamics of the Industrial Revolution and its anthropogenic emissions. The



upper section of CDD16-C01 can be dated using seasonal variations in chemical concentrations, and especially with $NH_4^+$ (Preunkert et al., 2000). Indeed, the typically high $NH_4^+$ concentrations found in summer reflect the vertical summer convection that transports $NH_4^+$ from lower elevations. Winters are associated with lower $NH_4^+$ concentration as the atmospheric boundary layer is well below the CDD site which limits the $NH_4^+$ supply from lower elevation sources.

As a proof of concept, we analysed $\delta^{15}N(NH_4^+)$ using the methodology presented here for the first 8 m w.e. at a resolution of
25 cm. One quarter of the ice core (102 mm diameter) was dedicated to $\delta^{15}N(NH_4^+)$ isotopic analyses, and this quarter was divided into two halves (named CDD16-C01-C and CDD16-C01-D) for reproducibility testing. Both of these reproducibility halves were analysed using the best methods approach determined by the previously described methodological tests, and reproducibility was examined by analysing the two sections by different operators and with both fresh and non-fresh cationic resin.

**4 Results and Discussion**

**4.1 Experimental $NH_4^+$ sample size**

The isotopic impact of $NH_4^+$ contamination from the blank can be considered negligible when changes in $NH_4^+$ sample size show no changes in $\delta^{15}N$ values. Based on our test replicating sample sizes from 20 to 100 nmol, we observe that $\delta^{15}N$ values are largely stable at sample sizes > 50 nmol (Figure 2). Conversely, for sample sizes below 30 nmol, the $NH_4^+$ coming from
the blank becomes a substantial proportion of the total $NH_4^+$ analysed per sample, with up to 46 % for 20 nmol samples. Unless it is possible to correct for the effects of the blank on the isotopic composition, 50 nmol is the smallest sample size that should be targeted while limiting isotopic effect from the blanks. However, the blank could also be corrected at the expense of greater isotopic value uncertainty, with the main limiting point being the acceptable level of uncertainty. This point will be further discussed in section 4.5.

The tests on three resin volumes (0.3, 0.6, and 1.2 mL) reveal no fractionation linked to the resin volume choice, assuming 100 % elution of the sample. Indeed, for standards J, K, and L, the non-corrected $\delta^{15}N$ values for the three volumes are (-6.8 ± 0.1) ‰, (-18.5 ± 0.1) ‰, and (13.0 ± 0.1) ‰ with ± representing the repeatability variations. These variations are smaller than the chemical's method standard deviation of 0.3 ‰ (Zhang et al., 2007). We can confidently say that resin volume we used does not affect the isotopic composition, probably because all the $NH_4^+$ is extracted during the elution step. This aspect
is further discussed in section 4.3.





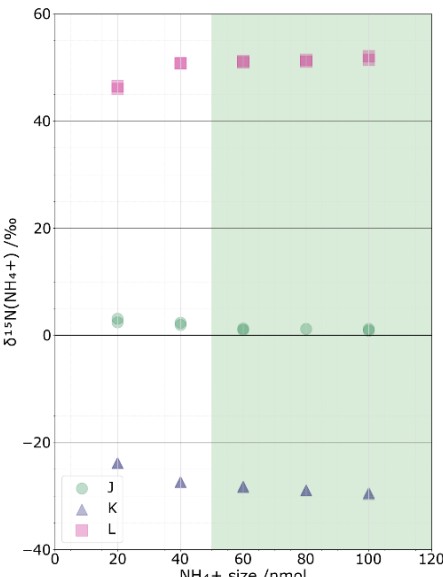

*Figure 2: $\delta^{15}N(NH_4^+)$ for three standards (‰) – J (red), K (green), and L (blue) – at different $NH_4^+$ sizes (nmol) and $\delta^{15}N$ linearity zone (faded green).*

## 4.2 Blank control

Three sources of contamination that contribute to the blank were considered: atmospheric $NH_3$ contamination, tool-induced contamination, and contamination during $NH_4^+$ conversion into nitrous oxide. Regarding the first contamination source, Figure 3 illustrates the relationship of $NH_4^+$ concentrations in the MQ-water vials kept open to the air for a certain amount of time under hoods with and without $NH_3$ scavenging system. Under a fume hood without a $NH_3$ scavenging system, MQ vials are immediately contaminated. For atmospheric $NH_3$ contamination, we found that a 5 mL MQ-water sample left for 2 hours in a

standard fume-hood reached 11 nmol $NH_4^+$ due to atmospheric $NH_3$ uptake. The use of $NH_3$ scavenging fume hood allows to reduce $NH_3$ contamination by a factor of $\approx 4$, reaching $\approx 2.8$ nmol $NH_4^+$ after 3 hours. While small, this $NH_3$ contamination level is still excessively high for ice core studies. This difficulty has been solved with the replacement of the $NH_3$ filter and its thickness that was halved to 5 cm. The previous filter was oversized hence resulting in a too high pressure drop inside the laminar flow, flowing too much external air in it. One should therefore check the contamination levels induced by atmospheric

$NH_3$ to adjust the size of the filter initially preinstalled. The contamination is now controlled to $0.26 \pm 0.02$ nmol from the atmospheric $NH_3$.



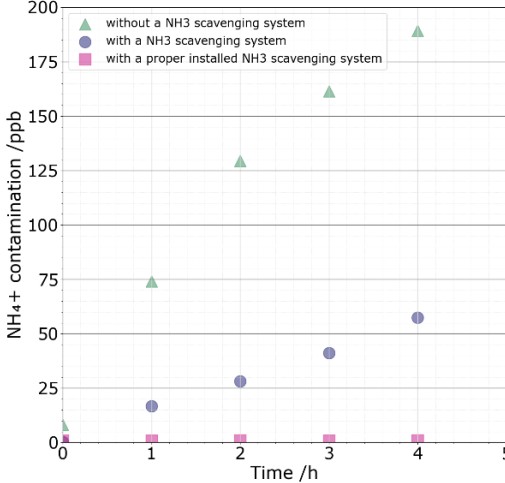

***Figure 3:*** *$NH_4^+$ contamination levels reported every hour under standard fume hood (green triangle), and under the $NH_3$ scavenging fume hood after installation (purple circle), and after changing the filter (pink square).*

Tool-induced contamination proved to be negligible. For the MQ ice stick which was decontaminated with non-precleaned tools, an $NH_4^+$ level of 0.70 ppb is reported. This level is decreased to 0.14 ppb when using the precleaned tools. This drop of 0.56 ppb would represent only < 1 % of a typical blank size of 13 nmol. Still, we recommend precleaning as a component of good practice as it takes little time.

The most significant contamination source was found to be associated with chemistry reactions and the $NaN_3$/acetic acid
solution in particular. Although the nitrite contamination from the reducing solution ($NaN_3$) was measured with Griess colorimetric dosage to be 27.5 ppb of $NO_2^-$ (which represents 0.2 nmol of contamination for the 0.5 mL used as reagent), this contamination is relatively insignificant compared to the 300 nmol $NH_4^+$ for a standard blank test of 5 mL of MQ water that was measured for the $NaN_3$/acetic acid solution. McIlvin & Altabet reported that this solution must be purged with He at 40 mL.min$^{-1}$ for 10 minutes (2005). However, in our case, we found that this blank was reduced to around 20 nmol after 10 min
of purging with He at 40 mL.min$^{-1}$, but that still remains too high for ice core analysis. We found that it was necessary to purge for at least 30 min to reach an acceptable value of 0.80 nmol. We suggest that the purging time is likely configuration-dependent, and this should be tested and set for each lab configuration.

Overall, our results suggest that contamination can be limited through meticulous planning and handling of samples. Summing all the contaminations previously mentioned leads to the blank size. Relative to 50 nmol samples, the overall contamination
present in the blank is still important, with an average 27 % size share relative to the sample. This requires that we develop a correction for the blank contribution through a mixing model (Section 4.5).

**4.3 5 mL of 1 M NaCl for a complete $NH_4^+$ elution**

Eluting each 50 nmol $NH_4^+$ sample with 0.1 M HCl, NaCl, and KOH achieved 113 %, 81 %, and 31 % sample recovery compared to the $NH_4^+$ sample reference that was not passed through the cationic resin. The < 100 % recovery for NaCl and




KOH indicates that some of the sample $NH_4^+$ was left in the resin or otherwise lost, while the $> 100$ % recovery for HCl indicates that the sample was contaminated with an external source of $NH_4^+$. These results also highlight the important role of pH and ion affinity in selecting the eluent. Since $K^+$ has a poorer affinity for the resin than $NH_4^+$, it is not unexpected that $NH_4^+$ could be left in the resin after the elution. Additionally, for the 0.1 M KOH, the pH (=13) is higher than pKa($NH_4^+$/$NH_3$), and thus the conversion of $NH_4^+$ into $NH_3$ is favoured. As a result, some sample $NH_4^+$ will be lost into the atmosphere as $NH_3$.

Both explanations, although we do not know which, make the KOH elution unsuitable in our case. For the 0.1 M HCl elution, the pH = 1 and the >100 % $NH_4^+$ recovery suggests that the acidic conditions are fixing atmospheric $NH_3$. In addition to this $NH_3$ fixation, HCl was reported to affect kinetics of $NH_4^+$ conversion into $NO_2^-$ (McIlvin and Altabet, 2005), and therefore we reject HCl as a viable eluent.

    This leaves NaCl as the best remaining eluent option, despite the only 81% recovery at 0.1 M. The conversion into $N_2O$ has
not been reported due to an NaCl matrix (McIlvin and Altabet, 2005), and so we assume that the low recovery of $NH_4^+$ is due to an uncomplete elution that is improved by simply increasing the NaCl eluent concentration.

    As the 81 % recovery from 0.1 M NaCl matrix is insufficient, we next tested a 1 M NaCl solution to determine if a higher concentration could more completely push out all the $NH_4^+$ ions. This test was comprised of 2x2.5 mL sequential elution fractions of 1 M NaCl, with each subsequent 5 mL eluent then analysed for $NH_4^+$ concentration (Figure 4). We found that 51.7
$\pm$ 1.9 nmol of $NH_4^+$ (1$\sigma$) were eluted after the first 5 mL elution. The second elution contained a very low 0.70 $\pm$ 0.59 nmol of $NH_4^+$, which is unlikely to significantly alter the isotopic composition. The subsequent fractions do not contain measurable $NH_4^+$. Based on these results, we selected our eluent to be 2x2.5 mL of 1 M NaCl.

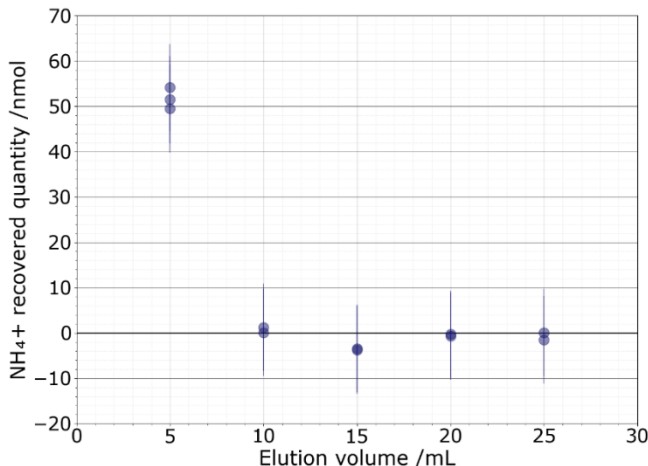

***Figure 4:*** *50 nmol $NH_4^+$ recovered quantity after 5 mL elution fractions with measurement uncertainty shown by vertical bars.*

**4.4 Isotopic and size correction with mixing model**

In contrast to a correction based on simple linear regression which requires identical sample amounts of $NH_4^+$ for standards and samples, our mixing model allows us to correct the method blanks and calibrate the samples. Figure 5 shows the results




of a correction applied to a series of standards. For this purpose, the international standards are prepared at different size (20 to 100 nmol) and process as described above (i.e., chemical conversion and IRMS analysis). Then using the mixing model approach and assuming that each analysis shares the same blank (amount and isotopic composition) across the different standards for a given batch, the python script calculates the shape of the mixing lines that best minimizes the difference between the modelled mixing lines and the standard points while converging toward the same origin, i.e., same blank. The best mixing line that passes through an unknown sample analysis (crosses in Figure 5b) is then used as a calibration curve to obtained the corrected isotope composition. Figure 5a shows the IRMS linear response and variability with respect to the amount of $NH_4^+$ while Figure 5b illustrates the mixing model curves (grey lines), the modelled blank and a simulated set of unknown samples (black crosses) made from the same reference materials that are used for calibration for illustration purpose. Figure 5c shows the residual between the model mixing curves and standards. The residual $\delta^{15}N$ of the standards is then used as a quantification of the precision of the method in the form of the standard deviation of the residual. With this approach, all standards of different sizes participate equally to the statistical correction of the method and allow to cover a broad range of isotopic compositions and sample sizes in one given batch of analysis (i.e., 4 blanks, up to 70 samples and 15 standards in our batch of analysis). In some cases, when the blank can be directly measured, it can be used as a post quality control point (grey dots of Figure 5a and 5b) by comparing its size and isotope composition with the modelled one.

Comparing a calibration from standards that passed through resins with one with standards that did not results in blank sizes of 5.4 Vs and 4.3 Vs, respectively, for the calibrations with resin-passed standards and non-resin-passed standards. The resin step increases the blank size by 20 %. An efficient mixing model is therefore needed to correct small size samples. Similarly, residual $\delta^{15}N$ increases from 0.41 ‰ to 0.59 ‰ in this set of analysis.



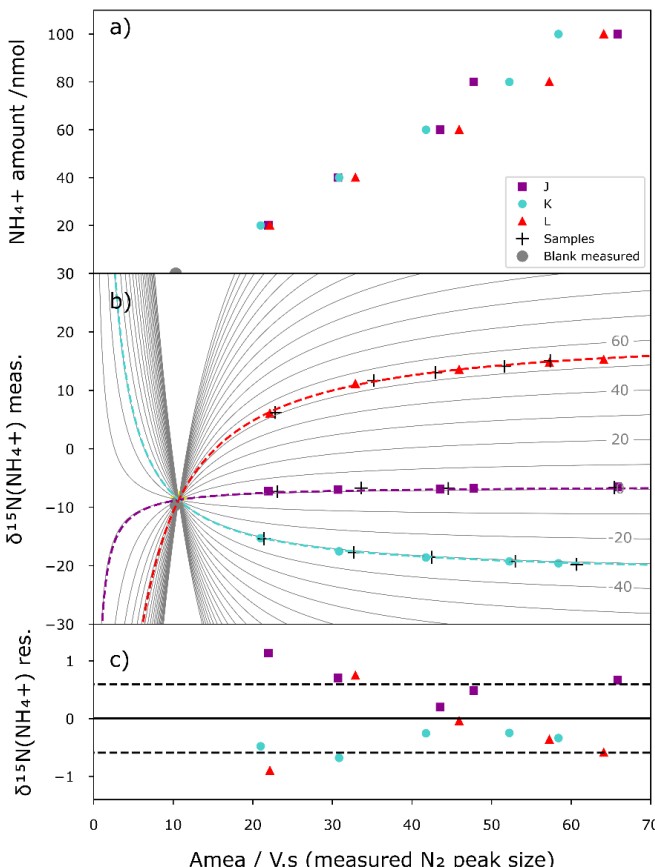

***Figure 5**: Mixing model correction. a) calibration curve between standard sizes (nmol) and IRMS peak areas (V.s) for three international isotopic standards J (green), K (blue), and L (red) and blanks (grey). b) mixing model curve fitting (dashed lines)*

*constrained by the measured $\delta^{15}N$ vs size of the standards (colour marks) and the measured blanks (grey dots). One set of standards are treated as unknown samples (black crosses) and corrected and calibrated with the mixing model (best fit of the continuous grey lines). c) Residual $\delta^{15}N$ from the difference between measured $\delta^{15}N$ of the standards and calculated $\delta^{15}N$ with the mixing model with 1σ standard deviation (dashed line)*

Figure 6 compares the mixing model-based correction with a simple linear regression correction for which we consider an

average calibration with respect to standard sizes. For the set of standards that were processed as unknown samples (black crosses Figure 5b), we quantify for both correction the difference between the corrected values and their known isotopic standard values. For the two corrections, the general trend is an increase of divergence between corrected and known values with smaller sizes. However, the mixing model-based correction performs better than a simple linear correction for all amounts tested. As expected, we notice a constant decline of the residual for the mixing model with respect to the $NH_4^+$ size. This

translates the increasing correction of the blank share for small sizes. For sample sizes > 40 nmol, the mixing model correction does not significantly improve the precision, showing that at this size level the blank becomes negligible, making the isotope




correction insensitive to sample size, as it should be if a blank component did not exist. In the same way, one can question the necessity of using a mixing model correction at sample sizes > 40 nmol as the difference between the linear correction and the international isotopic standard is not drastically different to each other. However, our approach demonstrates the usefulness of

the mixing model correction for small size samples. We believe that this should be valuable and incorporated in the data treatment for low concentrated environment such as polar ice cores or other environments with limited sample sizes such as meteorites and other extra-terrestrial samples. Before concluding this section, we want to emphasise that the 40 nmol threshold is laboratory-dependent and will highly depend on the blank level reached by each laboratory.

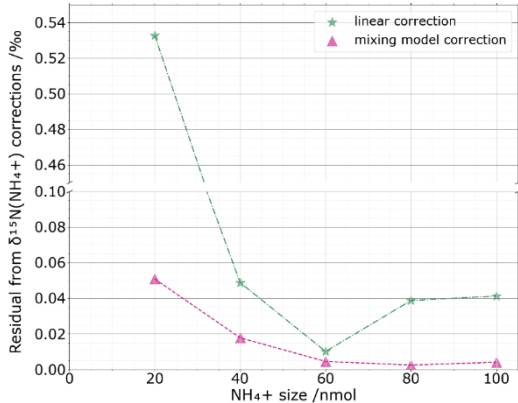

*Figure 6: For one set of standards processed as unknown samples, and for each sample NH₄⁺ size (nmol), we compare the absolute value of the residuals between the linear correction (green stars) and the mixing model correction (violet triangles) with the isotopic standard values.*

With the effect of the blank on the isotopic composition now under control (i.e., the residual < 0.2 ‰), samples with sizes < 50 nmol are only limited by the trade-off between size and acceptable uncertainty. The overall method we propose gives a

precision of ± 0.6 ‰ (1σ standard deviation) for samples greater than 20 nmol. This standard deviation is determined as the highest standard deviation for all our experiments calculated from the residuals of the mixing model. This ± 0.6 ‰ value can be compared with the ± 0.3 ‰ precision of Zhang et al.'s NH₄⁺ conversion (2007). Hence, although our proposed method has a higher standard deviation, it still offers the ability to measure $\delta^{15}N(NH_4^+)$ in low concentrated environment like ice cores with a relatively favourable standard deviation. To aptly validate the method, this standard deviation needs to be compared to

ice core $\delta^{15}N(NH_4^+)$ variability.

**4.5 Application of new methodology to CDD16-C01**

To test the applicability of our method, a portion of the CCD ice core was cut in two parallel sections with one used for NH₄⁺ concentration quantification and isotope analysis while the second was only used for isotope analysis reproducibility. The NH₄⁺ concentration profile measured on the continuous flow analysis system shows a well-marked seasonal pattern (Figure

7a), in agreement with a previous study (Preunkert et al., 2000). Using this seasonal NH₄⁺ concentration pattern, we date the



depth of 800 cm w.e. to be summer 2013. Isotopic values of $\delta^{15}N(NH_4^+)$ vary between (-17.7 ± 1.4) ‰ and (5.5 ± 1.4) ‰ across the two duplicated sections of the ice core CDD16-C01-C and CDD16-C01-D (Figure 7b). While the repeatability lies within ±2σ the method precision, the reproducibility between the two parallel isotopic analyses at 2σ is ± 2.1 ‰. Since the ice core's natural $\delta^{15}N(NH_4^+)$ values vary between -17.7 ‰ and 5.5 ‰, our analytical standard deviation of 0.6 ‰ and

reproducibility standard deviation of 2.1 ‰ shows that our overall proposed methodology has sufficiently low uncertainty for studying and interpreting ice core $NH_4^+$ isotope compositions.

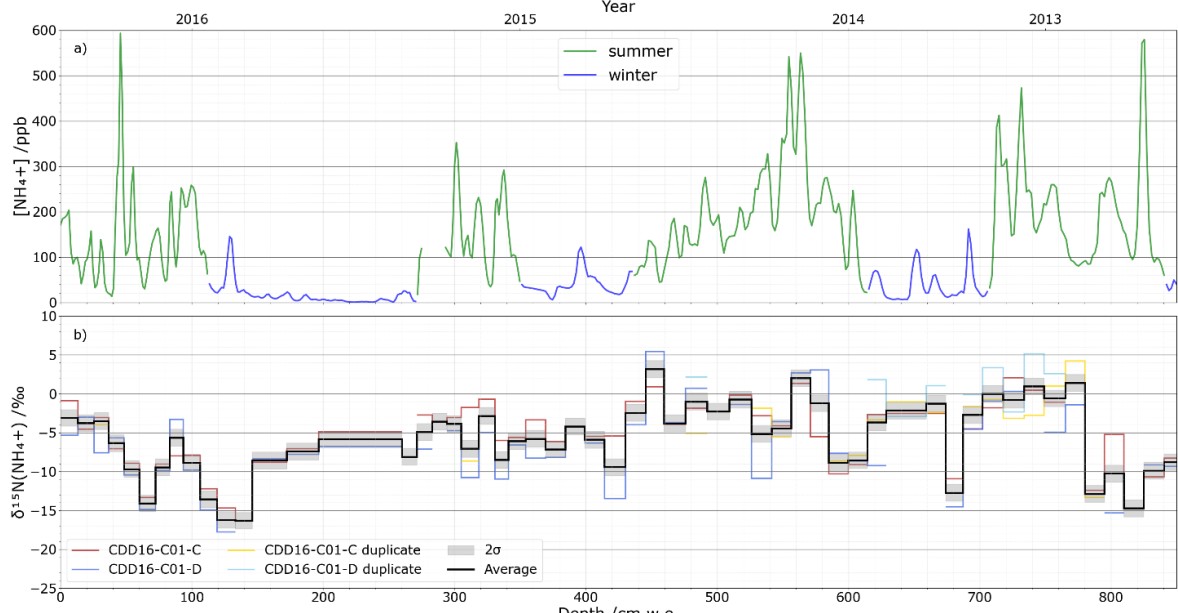

*Figure 7: a) Continuous flow analysis $NH_4^+$ concentration profile in ppb with summer (green) and winter (blue) from 2016 to 2013 (top). b) $\delta^{15}N(NH_4^+)$ isotopic profile in ‰ with the 25 cm ice sampling resolution for the two parallel cores CDD16-*

*C01-C (red) and CDD16-C01-D (blue) and their respective repeatability tests (yellow, light blue); the average value of all tests (black) is shown with the ± 2σ reproducibility.*

Although the environmental interpretation of this $NH_4^+$ isotopic record is not the focus of this paper, we briefly consider the potential drivers of the observed isotopic variability. The isotopic composition of $NH_4^+$ in an ice sample can be explained by the isotopic composition of the $NH_3$ emission sources and/or the isotopic fractionation that may take place in the atmosphere

or snow.

Because different $NH_3$ sources often have different ranges of $\delta^{15}N$ values, measuring $\delta^{15}N(NH_4^+)$ in ice cores offers the opportunity to investigate changes of $NH_3$ sources as long-term trends, seasonal variations, or one-off events. For example, agricultural sources have low $\delta^{15}N(NH_3)$ values ranging from -51.8 to -40.9 ‰ and -31.7 to -18.8 ‰ (Chang et al., 2016; Xiang et al., 2022), respectively and emissions from human and pet waste also leads to $^{15}N$-depleted $NH_3$ with signatures between -

41.9 and -29.9 ‰(Chang et al., 2019). On the other hand, higher $\delta^{15}N$ values might suggest greater contribution from



combustion sources from the industry (-14.6 to -11.3 ‰) or the automobiles (4.5 to 8.7 ‰)(Walters et al., 2020). Ice cores from sites such as the CDD should record seasonal variations in local and/or regional $NH_3$ sources. One study of the $\delta^{15}N(NH_4^+)$ composition of atmospheric particles in Lyon (150 km from the Mt-Blanc) has shown that the pollution peaks of ammonium nitrate in spring were accompanied by a strong decrease in $\delta^{15}N$ (Favez et al., 2021), potentially explained by agricultural

sources (livestock or fertilizers). However, this and other studies highlight the lack of broad understanding of isotopic variations over the rest of the year and difficulties in assigning diagnostic isotopic labelling to different emission pools (Mariappan et al., 2009; Felix et al., 2017; Favez et al., 2021).

Individual climate or environmental events may also leave distinct signals in an ice core's $NH_4^+$ record. In the Alps, for example, Saharan dust events are an important source of $NH_4^+$, and a single event can deliver high amounts of $NH_4^+$ within a

brief period. We captured one such dust event in the CDD ice core at 820 cm w.e. where the $NH_4^+$ concentration reaches its maximum value of nearly 600 ppb (Figure 7a). Based on our chronology and supported by CFA data from terrigenous species ($Mg^{2+}$, $Ca^{2+}$, $Al^{3+}$ etc.) and dust concentration, this $NH_4^+$ peak corresponds to a dust event on 30 April 2013. While it is unclear at this time whether such Saharan dust events have isotopic impacts (Figure 7b), our method offers a sampling resolution and isotopic precision fine enough to investigate these individual events.

The reactivity of $NH_4^+$ in the atmosphere can also cause isotopic fractionation after $NH_3$ has been emitted from its initial sources (Felix et al., 2017). $NH_3$ in the presence of acids, especially sulphuric acid ($H_2SO_4$), can form a particle nucleus (Kürten, 2019). Subsequently, the condensation of weak volatile acids such as $H_2SO_4$ and nitric acid ($HNO_3$) (Stolzenburg et al., 2020) with stabilisation by ammonia or organic bases (Lehtipalo et al., 2016) leads the particle into a growth phase. Walters et al. point out that kinetic isotope fractionation can take place during nucleation, whereas equilibrium fractionation occurs

during the growth phase (2019). In this regard, variations in the concentrations of the reagents could therefore modify these two fractionations. Assuming this process would not discriminate between different emission sources – i.e., the fractionation equally impacts all sources –, long term acidity trends could be investigated. Temperature can also affect atmospheric isotope fractionation (Li et al., 2012; Savard et al., 2017). The first results presented in Figure 7b do not allow us to conclude on the kinetic or equilibrium fractionations. Concerning temperature, however, as we do not observe any major difference between

the $\delta^{15}N$ values measured in summer and winter (Figure 7b), it seems unlikely that temperature plays a significant role in the fractionation. This aspect remains to be confirmed on a longer time scale.

Finally, isotopic fractionation is also possible after deposition within the snowpack through deprotonation of buried $NH_4^+$ and its re-volatilisation as $NH_3$. These post-depositional processes function analogously to the better documented post-deposition effects of nitrate in snow (e.g., Erbland et al., 2013). Strong $^{15}N$ enrichment would alter the overall signal remaining in the

snow as the isotopic fractionation constant $NH_4^+{}_{(s)} \leftrightarrow NH_{3(g)}$ is $\varepsilon = 31$ ‰ (Walters et al., 2019). However, these post-depositional processes are most impactful in the uppermost snow layers near the surface. Because CDD is a site of high accumulation (~2.5 m w.e. $yr^{-1}$) (Maupetit et al., 1995), any deposited $NH_4^+$ will be rapidly buried to depths where post-depositional phenomena are unlikely. Furthermore, over the first 60 cm w.e. of the $\delta^{15}N(NH_4^+)$ signal (Figure 7b), a decrease in $\delta^{15}N$ is recorded with no change of $[NH_4^+]$. This decrease is opposite to the trend that should be observed with an enrichment



of the snow in heavy isotopes if $NH_3$ were to be reemitted, besides a depletion in $[NH_4^+]$ should have been measured. We can therefore reject that post-depositional processes have a substantial effect on $NH_4^+$ isotopic values in the CDD snow. This question obviously remains to be investigated for sites marked by lower accumulations (e.g., East Antarctica highland).

## 5 Conclusion

Twenty-five years ago, Legrand and Mayewski stated that "the transfer functions of gaseous species that interact strongly with
ice are the most complex" and improving our analytical capabilities regarding this concept should be seen as one of the priorities of the ice core community (Legrand and Mayewski, 1997). New routes for studying $NH_3$, one of the most abundant species in inorganic aerosols, and its neutralised form $NH_4^+$ are therefore valuable and needed. This study presents a method for extracting $NH_4^+$ and analysing its $^{15}N$ isotopic composition in ice cores. The method hinges on three meticulous steps: a workflow under clean conditions, a cationic exchange resin with 1 M NaCl solution as the eluent to recover <50 nmol of $NH_4^+$,
and its conversion into a suitable gas for $\delta^{15}N(NH_4^+)$ analysis. The overall standard deviation of 0.6 ‰ and its reproducibility of 2.1 ‰ in an ice core from the Col Du Dôme (Mont-Blanc, France) are both much smaller than natural $^{15}N$ variability. These results validate our methodological approach, and we believe that our $NH_4^+$ analytical techniques can be helpful to reveal long term past atmospheric changes of $NH_3$ chemistry and emissions. Room for improvement still remains in the precision of the measurement, an issue that could be crucial for some sites where the range of $\delta^{15}N(NH_4^+)$ values would be lower than that
measured at CDD. We suggest that some additional precision could be gained by pre-drying the NaCl before preparing the eluent to eliminate possible contamination with moisture and flushing the vials with helium before chemical conversion to $NH_4^+$.

While $NH_3$ emissions have long been driven by natural soil emissions, the increasing anthropogenic impact should also be detectable in $\delta^{15}N(NH_4^+)$ in ice cores both in Alpine glaciers and polar ice sheets. As proposed controls and policies on $NH_3$
emissions are debated, ice core analysis of $\delta^{15}N(NH_4^+)$ from Alpine glaciers could allow an improved understanding of regional emission inventories. Looking to the future, the application of this method to Greenland ice could better constrain the changes and the origins of biomass burnings across the broader Northern Hemisphere (Rubino et al., 2016), while Antarctic $NH_4^+$ isotopic analysis may also be useful for exploring how marine biomass is affected by sea ice changes (e.g., Thomas et al., 2019). In these and other ways, our newly refined methodology opens many new opportunities for atmospheric chemists and
glaciologists to better understand how $NH_3$ and $NH_4^+$ have responded over time to human and natural changes in emissions, climate, and atmospheric composition.

## Acknowledgement

We acknowledge the LabEx OSUG@2020 ("Investissements d'avenir" – ANR10 LABX56), the ANR project ANR-15-IDEX-02 (IDEX-UGA), the IceMemory Foundation, and the technical support from the F2G (French National platform for Coring



and Drilling, handled by INSU, the MIUR (Ministry of Education, University and Research) for supporting this research. We also thank Sarah Albertin for scientific advice, Selin Bagci for lab handling, and Sophie Darfeuil for CFA colorimetric module development.

## Competing interests

The authors do not declare any conflict of interests.

**Code availability**

The scripts are available here https://doi.org/10.5281/zenodo.7728983 (Lamothe et al., 2023)

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
