# Peer review of "An extraction method for nitrogen isotope measurement of ammonium in low concentrated environment"

_EGUsphere, 2023_

## Author Response (AR1)

**Review 1**

This manuscript documents a detailed procedure for isotopic analysis of low-ammonium ice core samples. The experimental design is very-well thought and thorough, and the manuscript is well written and organized. All my questions about the work are answered within the manuscript, so I recommend publishing this manuscript as is in its current form.

**Answer**

We deeply thank reviewer 1 for his/her supportive comments. We are glad they appreciated the manuscript.

**Review 2**

The authors present an interesting manuscript on developing a methodology that can be useful for the measurement of nitrogen stable isotopes of low-concentration ammonium samples with implications for ice core studies. This work describes the development of a useful method with a focus on minimizing blank contamination and the concentration of low-concentration ammonium samples using a cation exchange resin. This work is timely and potentially impactful due to recent developments that have enabled the ability to measure low amounts of ammonium for isotope determination and will expand the community's ability to conduct these measurements for low-concentration samples. Overall, I recommend this manuscript for publication. I have only a few comments/suggestions that should be addressed.

We deeply thank reviewer 2 for his/her thorough reading and the detailed comments they provided. We are glad they found the paper of interest and hope we will satisfactorily address all the questions raised.

**Comments:**

**Figure 1:** The clarity of this figure should be improved. Specifically, for the "Chemical Conversion & IRMS" panel, the conversion of $NO_2^-$ to $N_2O$ is better represented using $HN_3$ and not $NaN_3$. Also, it would have been helpful if the m/z that the IRMS measures were indicated in the figure or at least the target molecule of analysis by the IRMS should be more clearly indicated. I can see that the $N_2O$ product is passed through an Au tube, which leads to $N_2$. Can the $N_2$ product also be highlighted in the panel for others to better understand the chain of events for $d15N$ isotope quantification? For the "Size correction and calibration" panel, it is not clear what the colors and lines represent. Also, $d15N(NH_4)$ is not being measured directly (you are measuring $d15N$ of product $N_2$), and this axis should be changed to $d15N(N_2)$ or $d15N(NH_4)$calibrated if this represents this calibrated value.

This Figure is central and can help one reader to properly understand the method. We therefore very welcome all the comments and edits the reviewer provided.

The quality of the figure has been improved in line with recommendations.

$HN_3$ now replaces $NaN_3$

m/z ratios were added as well as $N_2$

The subfigure 'size correction and calibration' is the same as figure 5. We acknowledge that no clear legend is present on the figure. We made the following changes:

- Added legend
- Changed the colours to make it colour-blind friendly
- Wrote $\delta^{15}N(N_2)$ instead of $\delta^{15}N(NH_4^+)$

We also make this last change for Figure 5

**Lines 117-118:** Kaiser et al., 2007 did not describe the measurement of d15N(NH4+). I think the authors reference Kaiser et al., 2007 due to the thermal decomposition of N2O to N2 for m/z analysis at 28, 29. The authors should clarify this point. Further, d15N(NH4+) is not measured directly, rather NH4+ samples are converted to N2 and the IRM measures m/z 28 and 29 for d15N determination and NH4+ unknowns are calibrated with respect to NH4+ reference materials.

The reviewer is absolutely right in the sense that Kaiser et al., 2007 present an online method to measure the isotopic composition of $NO_3^-$ thanks to the conversion into $N_2O$ and then into $N_2$. We note that our sentence was probably misleading and change it into:

"After $NH_4^+$ is converted into $N_2O$, the latter follows an online method (Kaiser et al. 2007) to be reduced into $N_2$ whose $\delta^{15}N$ isotopic composition is measured with a Finnigan MAT-253 IRMS."

**Lines 135-136:** The general reproducibility of the NH4+ concentration measurements should be reported here.

Thank you for this. We correct the sentence to consider this point:

"The overall system is suitable for ice core measurements with a limit of detection of $(0.22 \pm 0.03)$ ppb, a linear calibration down to 1 ppm, and a repeatability $<0.7$ %."

**Lines 144-145:** The details on the IC measurements, such as column types, eluent, and concentration, should be described.

We add these much needed details

"$NH_4^+$ concentrations for these contamination tests were measured through ion chromatography device (IC) device (ThermoFisher INTEGRION), connected to an autosampler (ThermoFisher AS/AP), with separation on CG16-4 µm and CS16-4 µm 2 mm columns. The mobile phase is an isocratic manually mixed MSA 29 mM solution for the

cations, applying a 0.16 mL min$^{-1}$ flow rate resulting in a 27-minute separation run with autoregenerated suppression and conductivity detection."

**Lines 148-149:** How were the MQ ice sticks made?

We thank the author for the pertinent question, and we have clarified the original sentence from:

"To investigate this route of contamination, we created artificial ice sticks made from MQ-pure water and replicated our typical ice core cutting and handling process."

to

"To investigate this route of contamination, we created artificial ice sticks made from cutting a block of frozen MQ-pure water and replicated our typical ice core cutting and handling process."

**Lines 180-181:** How might the presence of other cations that could also bind on the exchange resin impact the choice of resin volume? Is the final chosen resin volume (0.6 mL) valid for complete NH4+ capture and retention when flowing a large volume of solution with other cations present in solution?

Thank you for these questions to which we initially have also considered. The capacity of the resin is 2.1 mEq mL$^{-1}$. Since our method focuses on water and ice samples from environments with low ion concentrations, this volume of resin will always surpass its saturation by other cations. Additionally, we remind that the selectivity of $NH_4^+$ to the resin is higher compared to all cations except $H^+$, $Li^+$, and $Na^+$. Therefore, as our results show (Lines 280-285) the volume does not influence the isotopic composition. However, we would temper this statement by pointing out that these tests were carried out in environments with low concentrations (ppb range) and the question should therefore be studied for more concentrated environments (seawater, for example).

**Lines 204-207**: More information on the conversion of NH4+ to NO2- could be useful, because a large reagent blank was identified. How much of the hypobromite solution was added to each sample/standard? How long did the samples react with hypobromite? Was sodium arsenite added to each sample after oxidation? Further, for environmental samples (e.g., ice cores), which undoubtedly have more complex sample matrices, is the oxidation of NO2- to NH4+ always compete? Is the produced N2O peak areas from sample conversion near the expected value? Perhaps the python code could flag samples that don't produce N2O peak areas near the expectation based on the NH4+ concentration.

We understand the necessity for further details about the chemical conversion. We will therefore modify our paragraph accordingly:

"The measurement of $\delta^{15}N(NH_4^+)$ is performed according to the methods described by Zhang et al. (2007) and McIlvin & Altabet (2005). In details, 0.4 mL of hypobromite solution is added to the $NH_4^+$-containing vial closed with septum and the reactions is run for 30 min while agitating. The hypobromite solution is prepared in a container rinsed with MQ water, into which 20 mL of MQ water is added, followed by 1 mL of a bromate/bromide solution (0.6 g of $NaBrO_3$ + 5 g of NaBr in 250 mL of MQ water), and then 1.05 mL of 6 M

hydrochloric acid immediately after which the reaction is left for 5 minutes in the dark before the final addition of 20 mL of 10 M sodium hydroxide solution. Then, the excess hypobromite from its reaction with NH4+ is quenched adding 0.1 mL of an arsenide solution (5.1 g of NaAsO$_2$ in 100 mL MQ-water). The solution is let to stir for 1 minute. In the same vial, 1 mL of azide/acetic acid solution (1:1 v/v) previously flushed with Helium is added and the reduction is run for 30 min before the final addition of 1 mL NaOH 10 M to quench the excessing reagent."

Regarding the question whether the oxidation of NH4+ is complete, following Zhang et al (2007) results, a 30-minute reaction was reported to be optimal for MQ-water and seawater samples, which we assume the former to be close to ice core matrix.

As wisely suggested by the reviewer, the size difference between what is measured with the IRMS and what was expected from sample preparation is a way to flag suspicious samples. We note a 6 % difference between these two sizes (expected size vs IRMS size) for the tests we conducted to develop and validate the method, suggesting our method to be reliable. When it comes to the ice core example (section 4.5), the difference is larger between the size calculated from the ice core and what is measured with the IRMS (26 %). This difference is due to the calculation of the sample size (n(NH4+) = Concentration × Section × Length × density / Molar mass). Indeed, the fact to consider ice cores as perfect cylinders does not take into account the imperfections of the ice cores. Therefore a 7 % difference exists between the estimated density and the modelled density. This 7 % difference results in a 28 % difference for the sample sizes, in agreement with the 26 % difference. The difference between the IRMS and the CFA measurements is solely explained by calculation assumptions of the sample size. This also stresses the necessity to have a size correction model.

Because of the previously stated reasons, the nitrogen isotopic peak size is not an ideal precise marker of suspicious samples. However, we instead chose to flag samples whose sizes were extremely different (>2σ) from the mean peak size and followed up with examining the N/O peak ratio (since the oxygen isotopic composition is also measured for N2O), which is seen as a good marker of the accuracy of the measurement.

**Lines 309-310:** What about the blank associated with NH4+ conversion to NO2- (in particular, the hypobromite solution)? I'm a little confused by the description of the methodology if NH4+ is oxidized to NO2- in separate reaction vials and then transferred to a separate vial for conversion to N2O (as described in Zhang et al., 2006) or if the chemical conversions are happening in a single reaction vial.

We thank the reviewer for proposing an explanation of the large blank size. We try here to list the reasons why this step is unlikely to contribute to the blank:

- The hypobromite solution could be a source of contamination if in contact with atmospheric NH$_3$. This is why we prepare the hypobromite solution under the NH3-scavenging fume hood.
- NO2- present in the analyte can alter the isotopic composition as suggested in Zhang et al 2007 which we assume the reviewer is referring to. Since we perform a cationic solid phase extraction, all anions present in the analyte (including NO2-) are removed and should therefore not alter the isotopic composition

We thank the reviewer for highlighting the necessity to clarify the chemical conversion protocol and we hope that the answer to the previous question also help to address his/her question whether the reaction is run in single vial or separate vials.

**Lines 475-478:** Should this be "before chemical conversion to N2O" rather than "NH4"?

Absolutely! Thank you for pointing this out. It will be corrected.